# Protective mucosal immunity against SARS-CoV-2 after heterologous systemic prime-mucosal boost immunization

Dennis Lapuente [1✉], Jana Fuchs[1], Jonas Willar[1], Ana Vieira Antão[1], Valentina Eberlein[2,3], Nadja Uhlig [2,3], Leila Issmail[2,3], Anna Schmidt[1], Friederike Oltmanns[1], Antonia Sophia Peter[1], Sandra Mueller-Schmucker[1], Pascal Irrgang[1], Kirsten Fraedrich[1], Andrea Cara [4], Markus Hoffmann [5,6], Stefan Pöhlmann [5,6], Armin Ensser[1], Cordula Pertl[7], Torsten Willert[7], Christian Thirion[7], Thomas Grunwald[2,3], Klaus Überla[1] & Matthias Tenbusch [1✉]

Several effective SARS-CoV-2 vaccines are currently in use, but effective boosters are needed to maintain or increase immunity due to waning responses and the emergence of novel variants. Here we report that intranasal vaccinations with adenovirus 5 and 19a vectored vaccines following a systemic plasmid DNA or mRNA priming result in systemic and mucosal immunity in mice. In contrast to two intramuscular applications of an mRNA vaccine, intranasal boosts with adenoviral vectors induce high levels of mucosal IgA and lung-resident memory T cells ($T_{RM}$); mucosal neutralization of virus variants of concern is also enhanced. The mRNA prime provokes a comprehensive T cell response consisting of circulating and lung $T_{RM}$ after the boost, while the plasmid DNA prime induces mostly mucosal T cells. Concomitantly, the intranasal boost strategies lead to complete protection against a SARS-CoV-2 infection in mice. Our data thus suggest that mucosal booster immunizations after mRNA priming is a promising approach to establish mucosal immunity in addition to systemic responses.

[1] Institute of Clinical and Molecular Virology, University Hospital Erlangen, Friedrich-Alexander University Erlangen-Nürnberg, Erlangen, Germany. [2] Department of Immunology, Fraunhofer Institute for Cell Therapy and Immunology, IZI, Leipzig, Germany. [3] Fraunhofer Cluster of Excellence Immune-mediated Diseases CIMD, Frankfurt am Main, Germany. [4] National Center for Global Health, Istituto Superiore di Sanità, Rome, Italy. [5] Infection Biology Unit, German Primate Center-Leibniz Institute for Primate Research, Göttingen, Germany. [6] Faculty of Biology and Psychology, Georg-August-University Göttingen, Göttingen, Germany. [7] Sirion Biotech, Martinsried, Germany. ✉email: Dennis.Lapuente@uk-erlangen.de; matthias.tenbusch@fau.de

The severe acute respiratory syndrome coronavirus 2 (SARS-CoV-2) emerged in late 2019 and caused a world-wide pandemic accounting for over 190 million infections and 4 million deaths at the time of this report[1]. In an unprecedented speed, academic institutions and biotech companies developed, evaluated, and licensed several SARS-CoV-2 vaccines. Beside traditional approaches like protein subunit or inactivated virus vaccines, gene-based vaccines were at the forefront of the developmental process and the first to become licensed[2].

Vaccines based on messenger RNA (mRNA) or adenoviral vectors (Ad) demonstrated efficacy against SARS-CoV-2 infections and, most importantly, against severe coronavirus disease 2019 (COVID-19) and death[3-6]. Humoral as well as cellular immune responses against the spike (S) surface protein were successfully induced by both types of vaccines[7-11]. However, breakthrough infections of fully vaccinated individuals have been reported and the numbers might increase in the phase of waning immunity[12-18]. The impact of immune escape and newly emerging virus variants (i.e. variants of concern, VOCs) is controversially discussed in some of these studies. Upon breakthrough infections, virus replication in the respiratory tract is approximately four- to six-fold reduced compared to unvaccinated and virus shedding seems to be shorter in duration[14,19]. Importantly, Public Health England reported that after the first dose of an mRNA (Comirnaty®) or viral vector vaccine (Vaxzevria®) the likelihood of household transmission drops by 40–50%[20]. On one hand, these observations underline that the current vaccination campaigns can end the pandemic phase by reducing the basic reproduction number below 1. On the other hand, however, it also demonstrates that transmission is still possible by vaccinated individuals posing a risk to vulnerable communities.

While the currently approved vaccines induce systemic immune responses, they probably do not evoke mucosal immunity in form of mucosal, secretory immunoglobulin A (IgA) or tissue-resident memory T cells ($T_{RM}$). Secretory, polymeric IgA can neutralize incoming viral particles at the mucosal surface before infection of epithelial cells takes place, which is important for an optimal protection against respiratory virus infections[21-23]. Furthermore, IgA enables specific effector functions by cross-linking the Fcα-receptor, and polymeric forms of IgA might increase antibody avidity[24]. So far, the only licenced intranasal vaccines are live-attenuated influenza vaccines (LAIV). Nasal IgA contributes to the efficacy of these vaccines in children[25] and also correlates with protection in experimental human challenge studies[26]. Importantly, local antigen deposition by mucosal vaccination routes is key for an induction of mucosal IgA as shown in humans[24,27-29] and animal models[30-32]. While IgA can be effectively induced by intranasal delivery of protein-based vaccines, an efficient induction of respiratory CD8+ $T_{RM}$ usually requires local antigen production in the mucosa followed by major histocompatibility complex-I-mediated peptide-presentation by stromal and, most importantly, by migratory CD103+ dendritic cells[33]. CD8+ $T_{RM}$ localize within the respiratory epithelium or the airways and can respond immediately in case of secondary infections. In contrast to circulatory T cell memory phenotypes like central memory ($T_{CM}$), effector-memory ($T_{EM}$), or effector T cells ($T_{EFF}$), $T_{RM}$ do not significantly recirculate[34,35]. Thus, one feature of $T_{RM}$ is the direct localization at barrier tissues, which makes a time-consuming migration into the inflamed lung redundant. A second remarkable characteristic is the ability to exert innate and adaptive functions within a few hours after secondary infection[36,37], in part due to the storage of ready-made mRNAs encoding cytokines like IFNγ at steady state[38,39]. Altogether, these unique features of mucosal immune responses enable an immediate and effective countermeasure

against pulmonary infections as described for flu[40,41], respiratory syncytial virus (RSV)[42], and *Mycobacterium tuberculosis*[43,44]. The great majority of these findings were generated in animal models, partly due to the invasive nature of bronchoalveolar lavages (BAL) and biopsy sampling. However, small experimental human challenge studies started to look precisely at the role of mucosal immunity against respiratory viruses[45,46].

A few preclinical studies investigated intranasal SARS-CoV-2 vaccines so far. In a series of publications, one group reported protective efficacy of a one shot vaccination with an chimpanzee adenoviral vector (ChAd-SARS-CoV-2-S) vaccine encoding for the spike protein in mice, hamsters, and rhesus macaques[47-49]. Importantly, van Doremalen et al. have shown that intramuscular adenoviral vector (ChAdOx1) vaccination prevents pneumonia in macaques but allow for virus replication in the upper respiratory tract[50]. However, administered intranasally, the vaccine attenuated nasal virus replication more efficiently[51]. It is important to investigate intranasal vaccine candidates not only as standalone modalities but also in the context of pre-existing immunity induced by a previous vaccination. On one hand, this is important due to the broad employment SARS-CoV-2 vaccines in recent vaccination campaigns. On the other hand, first clinical data point towards suboptimal immunogenicity of solely intranasal vaccinations against SARS-CoV-2 in humans without pre-existing immunity, but also provides evidence for robust immunity after heterologous prime-boost vaccinations[52,53].

In this study, mucosal vaccinations with adenoviral serotype 5 and 19a vectors are assessed in mice with or without prior systemic priming. We demonstrate that a systemic plasmid DNA or mRNA prime followed by an intranasal boost with an adenoviral serotype 5 vector (Ad5) enables a comprehensive systemic and local T cell immunity as well as substantial mucosal neutralization of SARS-CoV-2 VOCs. Concomitantly, the immunity established by mucosal boost strategies efficiently controls virus replication upon experimental infection comparable to homologous systemic immunizations. Considering mucosal boost immunization in individuals prior vaccinated with mRNA vaccines, may increase the efficacy of ongoing vaccination campaigns against SARS-CoV-2. Furthermore, the concept might be also transferable for the future management of newly emerging respiratory tract viruses.

## Results

**A systemic plasmid DNA prime significantly increases the mucosal immunogenicity of an intranasal adenoviral vector vaccine.** In this first part of our study, we evaluated the immunogenicity of mucosally applied viral vector vaccines as a single shot vaccine or as a booster after an intramuscular plasmid DNA prime immunization. To this end, codon-optimized sequences encoding the full-length S and nucleocapsid (N) proteins of SARS-CoV-2 were inserted into pVax-1 expression plasmids (in the following designated as plasmid DNA vaccine) and into replication-deficient adenoviral vector vaccines based on serotype 5 (Ad5) or serotype 19a (Ad19a). BALB/c mice were immunized intranasally with the Ad5- or Ad-19a-based vaccines either without prior treatment or four weeks after an intramuscular plasmid DNA immunization with S- and N-encoding plasmids (Fig. 1A). Two weeks later, SARS-CoV-2 specific antibody responses were analysed in serum and BAL samples, whereas the local and systemic T cell responses were determined in lungs and spleens, respectively.

In our flow cytometric assay[54], spike-specific IgG, IgG1, and IgG2a could be easily detected in serum and BAL of animals treated with the prime-boost strategies, while antibodies in the BAL after a single dose of Ad19a or Ad5 were almost absent

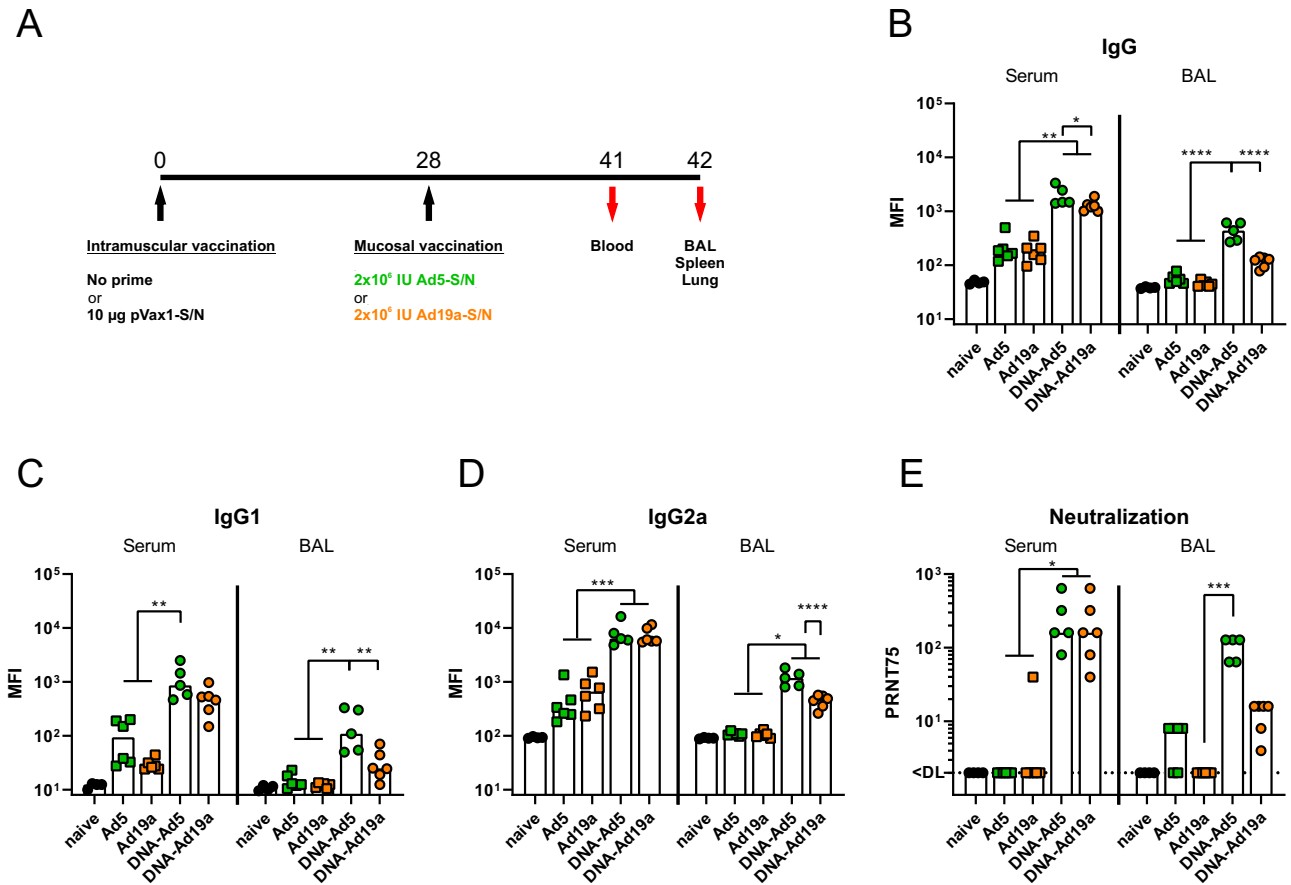

**Fig. 1 Humoral responses after intranasal immunization with Ad5- or Ad19a-based viral vector vaccines. A** BALB/c mice were immunized intranasally with Ad5- or Ad19a-based vectors encoding the N and S protein of SARS-CoV-2 ($2 \times 10^6$ infectious units per vector). Mice from the heterologous prime-boost groups were primed four weeks before by intramuscular injection of N- and S-encoding DNA plasmids (10 μg per plasmid) followed by electroporation. Serum antibody responses were analysed thirteen days and mucosal immune responses in the BALs fourteen days after the mucosal immunization. Spike-specific IgG (**B**), IgG1 (**C**), and IgG2a (**D**) were assessed by a flow cytometric approach (dilutions: sera 1:400, BAL 1:100). Plaque reduction neutralization titres (PRNT75) were determined by in vitro neutralization assays (**E**). Bars represent group medians overlaid with individual data points; naïve $n = 4$; DNA-Ad5 $n = 5$; other groups $n = 6$. Data were analysed by one-way ANOVA followed by Tukey's post test (**B–D**) or by Kruskal–Wallis test (one-way ANOVA) followed by Dunn's multiple comparison test (**E**). Statistically significant differences are indicated only among the different vaccine groups; $p$ values indicate significant differences (*$p < 0.05$; **$p < 0.005$; ***$p < 0.0005$; ****$p < 0.0001$).

(Fig. 1B–D). Comparing the two adenoviral vectors as booster vaccines, the serotype 5 induced significantly higher levels of S-specific antibodies in the BAL, although the antibody levels in the sera were comparable for the both groups. This effect was confirmed in the IgG subclass analyses for both, IgG1 and IgG2a levels (Fig. 1C, D). Similar trends were also observed for N-specific antibody levels in sera and BALs (Supplementary Fig. 1). In line with the amount of S-binding antibodies, profound virus neutralization was detected in sera and BAL samples from the groups DNA-Ad5 and DNA-Ad19a, while Ad5 or Ad19a alone did not induce significant levels of neutralizing antibodies (Fig. 1E). Given the differences in the local antibody levels, the IgA response in the BAL towards specific domains of the S protein were analysed in more detail by ELISA (Fig. 2A–C). These results confirmed that intranasal applications of Ad5-based vectors induce higher S-specific IgA levels than Ad19a-based vectors and these responses benefit from a systemic plasmid DNA prime. Furthermore, the vaccine-induced antibodies were directed against S1 including the receptor-binding domain (RBD) as well as against the S2 domain of the spike protein (Fig. 2A–C).

Next, we assessed the induction of cellular immune responses in the lung by the different vaccination schemes. Intravascular staining (iv-labelling)[55] was used to differentiate between

circulating T cells present in the lung endothelium during sampling (iv−) and $T_{RM}$ (iv+). Since specific MHC-I multimers were not available at the time of this study, antigen-experienced T cells were identified by the expression of CD44 (gating strategy shown in Supplementary Fig. 2). Similar to the humoral responses, CD44$^+$ CD8$^+$ T cells in the lung were most efficiently induced by the DNA-Ad5 scheme, although all treated animals mounted vaccine-induced cellular responses (Fig. 3A). The vast majority of lung CD8$^+$ T cells were protected from the iv-labelling in all groups, and the most prominent $T_{RM}$ phenotype was CD103$^+$CD69$^+$ (Fig. 3B). Antigen-specific CD4$^+$ and CD8$^+$ T cells were identified by ex vivo restimulation with peptide pools covering major parts of S and the complete N protein, respectively, followed by intracellular staining of accumulated cytokines (gating strategy in Supplementary Fig. 3). The highest percentages of S-reactive CD8$^+$ T cells were detected in the lungs of DNA-Ad5 treated animals with the majority of them predominantly producing IFNγ (Fig. 4A). Differences in the percentages of CD8$^+$ T cells expressing IL-2 or TNF were less pronounced, and polyfunctional T cells positive for all four analytes including the degranulation marker CD107a were rarely found in all animals. In contrast, significantly elevated percentages of CD8$^+$ T cells producing IFNγ or TNF as well as

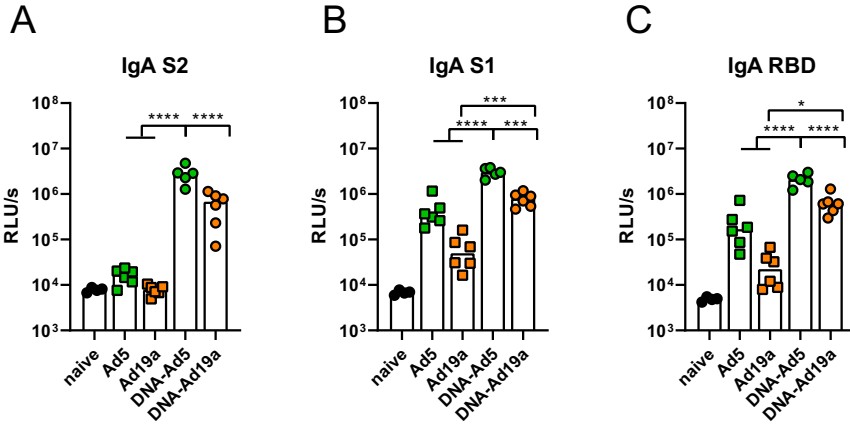

**Fig. 2 Mucosal, spike-specific IgA responses.** BALB/c mice were vaccinated according to Fig. 1A. BAL samples were tested for spike-specific IgA directed against the domains of S2 (**A**), S1 (**B**), or RBD (**C**) by ELISA (dilution: 1:10). Bars represent group medians overlaid with individual data points; naïve $n = 4$; DNA-Ad5 $n = 5$; other groups $n = 6$. Data were analysed by one-way ANOVA followed by Tukey's post test. Statistically significant differences are indicated only among the different vaccine groups; $p$ values indicate significant differences (*$p < 0.05$; **$p < 0.005$; ***$p < 0.0005$; ****$p < 0.0001$).

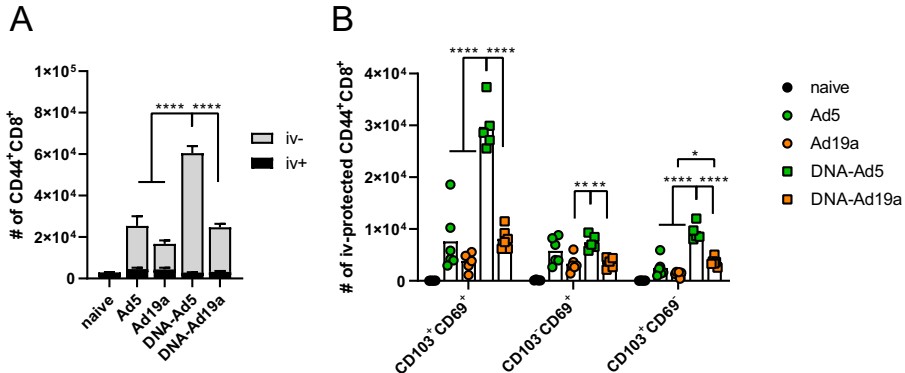

**Fig. 3 Tissue-resident memory T cell subsets in the lung.** BALB/c mice were vaccinated according to Fig. 1A. In absence of suitable MHC-I multimers, antigen-experienced CD8+ T cells were identified by CD44 staining (**A**). Intravascular staining was used to differentiate between circulating (iv+) and tissue-resident (iv−) memory cells. Tissue-resident phenotypes were assessed by staining for CD69 and/or CD103 within the iv-protected memory compartment (**B**). The gating strategy is shown in Supplementary Fig. 2. Bars represent group means with SEM (**A**) or overlaid with individual data points (**B**); naïve $n = 4$; DNA-Ad5 $n = 5$; other groups $n = 6$. Data were analysed by one-way ANOVA followed by Tukey's multiple comparison test. Statistical significant differences are indicated only among the different the vaccine groups; $p$ values indicate significant differences (*$p < 0.05$; **$p < 0.005$; ***$p < 0.0005$; ****$p < 0.0001$).

polyfunctional CD8+ T cells were detected in the spleens of DNA-Ad19a treated animals (Fig. 4C). Albeit at overall lower frequencies, the same observation was made for N-reactive CD8+ T cells in lungs and spleens (Supplementary Fig. 4A, C). Pronounced S- and N-specific CD4+ T cell responses were detected in all animals that received a prime-boost vaccination (Fig. 4 and Supplementary Fig. 4). In contrast to the CD8+ T cells, the majority of the CD4+ T cells were polyfunctional indicated by the simultaneous expression of IFNγ, TNF and IL-2. Again, immunization with the Ad19a-based vectors resulted in higher systemic responses measured in the spleen, whereas the mucosal response in the lung was more pronounced after delivery of Ad5-based vectors (Fig. 4C, D, Supplementary Fig. 4C, D).

Taken together, Ad5 proved a higher immunogenicity as mucosal vaccine vector compared to Ad19a and resulted in strong cellular and humoral immune responses against SARS-CoV-2 antigens if combined with an intramuscular plasmid DNA prime immunization.

**An intranasal boost following systemic mRNA vaccination potentiates mucosal antibody responses with pronounced neutralization breadth.** Since mRNA vaccines are currently in use for mass vaccination campaigns in many countries, we wanted to compare the differential effects of a plasmid DNA or mRNA prime on the immunogenicity of a mucosal booster. Therefore, the previously described DNA-Ad5 scheme was compared to an mRNA prime (Comirnaty®, Biontech/Pfizer) followed by an intranasal Ad5 boost (RNA-Ad5). Moreover, two vaccine groups that received two intramuscular injections with either mRNA (2x RNA) or an adenoviral vector (2x Ad5) reflecting current SARS-CoV-2 vaccination strategies were included (Fig. 5A). These experiments were performed in C57BL/6 mice to allow correlations to efficacy data in K18-hACE2 mice.

Four weeks after the boost immunization, all vaccinated animals reached high levels of anti-S IgG in the serum (Fig. 5B and Supplementary Fig. 5). However, the anti-S IgG levels after the homologous RNA vaccination were significantly higher than in all other groups. Interestingly, this order does not reflect the anti-S response measured four weeks after the prime immunization. Here, the intramuscular injection of Ad5 induced the highest antibody levels, most probably by inducing more potent IgG2a responses than the RNA vaccine (Supplementary Fig. 6). Contrary, the IgG levels detected in BALs were higher in the groups receiving the intranasal Ad5 boost vaccination (Fig. 5B).

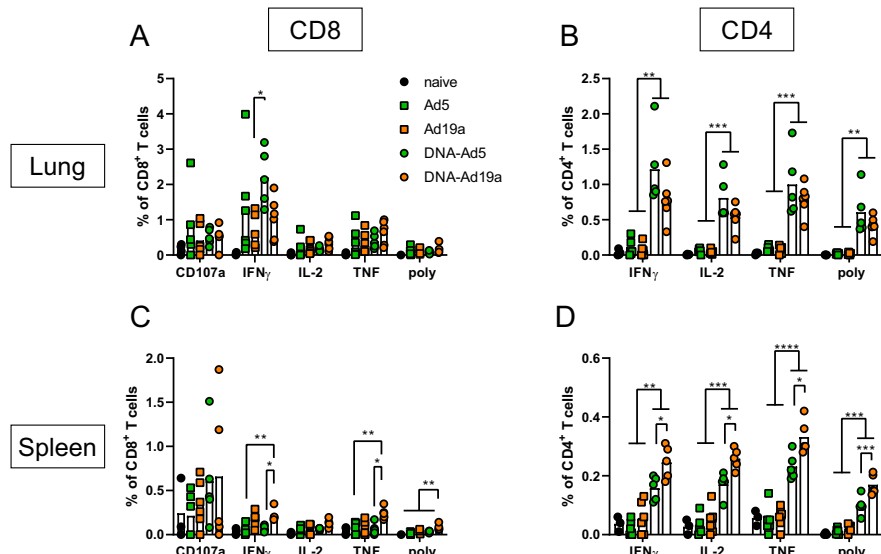

**Fig. 4 Spike-specific T cell responses after intranasal immunization with Ad5- or Ad19a-based viral vector vaccines.** BALB/c mice were vaccinated according to Fig. 1A. Lung and spleen homogenates were restimulated with peptide pools covering major parts of S. The responding CD8$^+$ (**A** and **C**) and CD4$^+$ T cells (**B** and **D**) were identified by intracellular staining for accumulated cytokines or staining for CD107a as degranulation marker. The gating strategy is shown in Supplementary Fig. 3. Bars represent group means overlaid with individual data points; naïve $n = 4$ (exception: $n = 3$ in **C** and **D**); DNA-Ad5 $n = 5$; other groups $n = 6$. Data were analysed by one-way ANOVA followed by Tukey's multiple comparison test. Statistically significant differences are indicated only among the different vaccine groups; $p$ values indicate significant differences (*$p < 0.05$; **$p < 0.005$; ***$p < 0.0005$; ****$p < 0.0001$). poly; polyfunctional T cell population positive for all assessed markers.

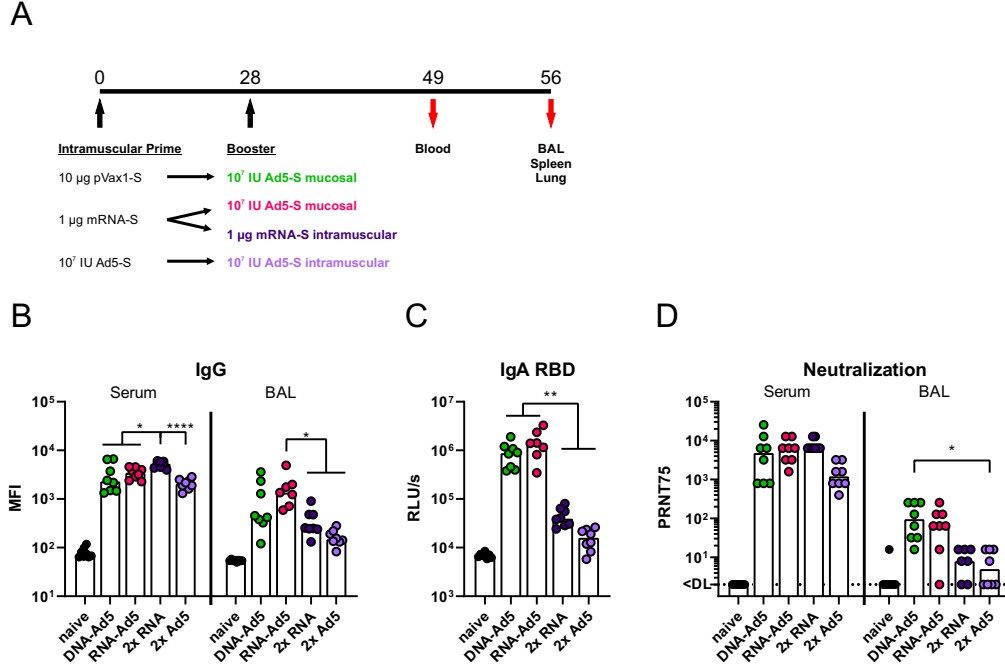

**Fig. 5 Humoral responses after homologous or heterologous prime-boost vaccination. A** C57BL/6 mice received an intramuscular prime immunization with the spike-encoding DNA (10 µg), Ad5-S (10$^7$ infectious units), or the mRNA vaccine, Comirnaty® (1 µg). Mice from the heterologous prime-boost groups were boosted four weeks later intranasally with Ad5-S (10$^7$ infectious units). The homologous prime-boost groups received a second dose of mRNA (1 µg) or Ad5-S (10$^7$ infectious units) intramuscularly. Serum antibody responses were analysed 21 days and mucosal immune responses four weeks after the boost immunizations. Spike-specific IgG (**B**) were assessed by a flow cytometric approach (dilutions: Sera 1:800, BAL 1:20). BAL samples were tested for spike-specific IgA directed against RBD by ELISA (**C**). Plaque reduction neutralization titres (PRNT75) were determined by in vitro neutralization assays (**D**). Bars represent group medians overlaid with individual data points; sera all groups $n = 8$; BALs RNA-Ad5 $n = 7$, other groups $n = 8$ (out of two independent experiments). Data were analysed by one-way ANOVA followed by Tukey's post test (**B** and **C**) or Kruskal–Wallis test (one-way ANOVA) followed by Dunn's multiple comparison test (**D**). Statistically significant differences are indicated only among the different vaccine groups; $p$ values indicate significant differences (*$p < 0.05$; **$p < 0.005$; ***$p < 0.0005$; ****$p < 0.0001$).

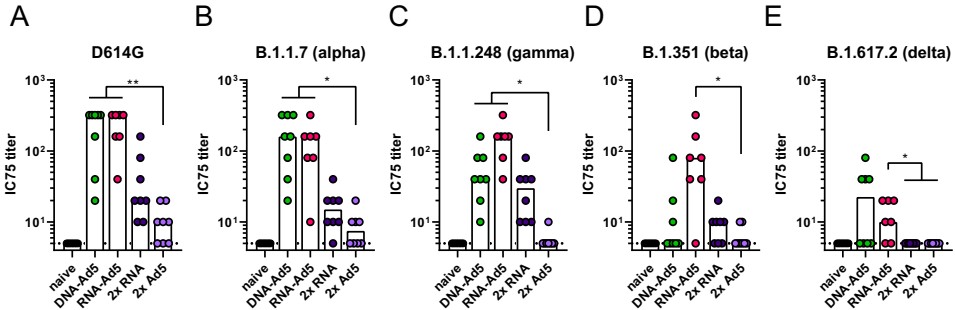

**Fig. 6 Neutralization of SARS-CoV-2 variants.** C57BL/6 mice were vaccinated according to Fig. 5A. BAL samples were analysed by pseudotype neutralization assays for the neutralization of different SARS-CoV-2 variants (**A–E**). Data points were shown for individual animals and bars represent group medians; RNA-Ad5 $n = 7$, other groups $n = 8$ (out of two independent experiments). The dashed line indicates the lower limit of detection. Data were analysed by Kruskal–Wallis test (one-way ANOVA) followed by Dunn's multiple comparison. Statistically significant differences are indicated only among the different vaccine groups; $p$ values indicate significant differences (*$p < 0.05$; **$p < 0.005$; ***$p < 0.0005$; ****$p < 0.0001$).

In addition, significantly increased local IgA antibody levels could be detected for both groups in a RBD-specific ELISA (Fig. 5C). On a functional level, the higher amounts of RBD-specific antibodies were mirrored by higher neutralizing capacities in the BALs of the groups DNA-Ad5 or RNA-Ad5 (Fig. 5D). Interestingly, the high amount of neutralizing antibodies in the sera were not significantly different among the vaccine groups independent of the route of the boost immunization.

Since mucosal antibodies might be most important for preventing an initial infection and thereby transmission, we evaluated the protective capacity against SARS-CoV-2 variants in pseudotype-based virus neutralization assays (Fig. 6). Here, the most robust and broadest responses were detected in the BALs of RNA-Ad5 treated animals with decreasing neutralizing potencies against spike proteins from SARS-CoV-2 lineages D614G to B.1.1.7 (alpha variant)/P.1 (gamma variant) to B.1.351 (beta variant), and finally B.1.617.2 (delta variant). Interestingly, the RNA-Ad5 and DNA-Ad5 schemes resulted in comparable IC75 titres against alpha and delta, but DNA-Ad5 was less potent against the beta variant. This might reflect the different nature of the encoded S protein sequences. Finally, the solely systemic vaccination schedules provoked 4- to 32-fold lower titres of mucosal neutralization against D614G, alpha, beta, and gamma, whereas no neutralization of delta spike-pseudotyped reporter virus could be observed. These data underline the importance of mucosal immunizations in order to provide immediate neutralization of incoming virus at the entry site.

**Lung-resident memory T cells are efficiently established by a mucosal boost but not by conventional mRNA vaccination.** Next, we assessed the induction of systemic and resident T cell memory. Antigen-experienced CD44$^+$ CD8$^+$ T cells isolated from lung tissue were quantitatively most pronounced in the 2x RNA group (Fig. 7B). However, by analysing the contribution of tissue-resident (iv−) and vascular (iv+) compartments, a more complex picture emerged. The groups that received two systemic immunizations almost exclusively mounted circulating T cell memory (>95% iv+; Fig. 7A, B) and consistent to this, the predominant memory phenotypes were T$_{EFF}$, T$_{EM}$, and T$_{CM}$ (Fig. 7C). CD103$^+$CD69$^+$ T$_{RM}$ were not established in the lungs of these animals. In complete contrast, the DNA-Ad5 immunized animals displayed mostly T$_{RM}$ but were lacking substantial numbers of circulating memory cells. Importantly, the RNA-Ad5 strategy induced the most comprehensive T cell memory consisting of both circulating subsets and CD103$^+$CD69$^+$ T cells in the lung.

The analysis of spike-specific, cytokine producing CD8$^+$ T cells showed a similar compartmentalization. Although the overall numbers of CD107a$^+$, IFNγ$^+$, and TNF$^+$ CD8$^+$ T cells were highest in the lungs of the 2x RNA group, these cells were almost exclusively found in the vascular compartment (iv-labelled, Fig. 8A–C). The same is true for the homologous immunization with Ad5, albeit reaching much lower percentages of reactive cells. In line with the phenotypic analyses, RNA-Ad5 induced both systemic and local T cell responses, whereas DNA-Ad5 provoked mainly T$_{RM}$. The trends observed for CD8$^+$ T cell responses in the iv-labelled lung population were largely mirrored by the splenic responses (Fig. 8D), further underlining that the former population reflects circulating T cells present in the lung vasculature at the time of sampling. Spike-specific, tissue-resident CD4$^+$ T cell responses were also effectively established by the mucosal boost strategies (Fig. 9A, B) and systemic IFNγ-producing CD4$^+$ T cells in the spleen were induced by all vaccine schedules with two RNA shots being the most effective strategy (Fig. 9D).

In conclusion, only intranasal vaccination schedules were able to induce profound mucosal immunity in the respiratory tract consisting of neutralizing IgG, IgA, and lung T$_{RM}$. Compared to DNA-Ad5, the RNA-Ad5 strategy provoked a more efficient neutralization of VOCs and established a comprehensive T cell immunity consisting of both T$_{RM}$ and circulatory T cells.

**Systemic and mucosal vaccine schedules effectively protect from experimental SARS-CoV-2 infection.** In order to assess the protective efficacy of the vaccination strategies, human ACE2 transgenic mice (K18-hACE2) were immunized as described before and challenged four weeks after the boost immunization with $9 \times 10^3$ FFU of the SARS-CoV-2 strain BavPat1 as previously described[56]. Since the 2x Ad5 immunization was less immunogenic than the 2x RNA immunization, this group was replaced by another 2x Ad vaccination regime consisting of an intramuscular Ad19a prime followed by the established intranasal Ad5 boost (Fig. 10A). Seven out of eight unvaccinated control animals reached humane endpoints at day five indicating a severe and lethal course of the disease (Fig. 10B). They presented weight loss starting at day four post-infection with a concomitant increase of clinical signs (Fig. 10C, D). In contrast, all vaccinated groups were largely protected from weight loss, clinical signs of disease, and mortality (Fig. 10B–D). High levels of viral RNA in lung homogenates and BAL fluids were only detected in unvaccinated animals indicating efficient viral replication, while from the vaccinated animals only two of the 2x RNA group had viral RNA copy numbers in the lung above the detection limit (Fig. 10E). Similarly, infectious virus was retrieved from the lungs of unvaccinated animals but not from the immunized groups

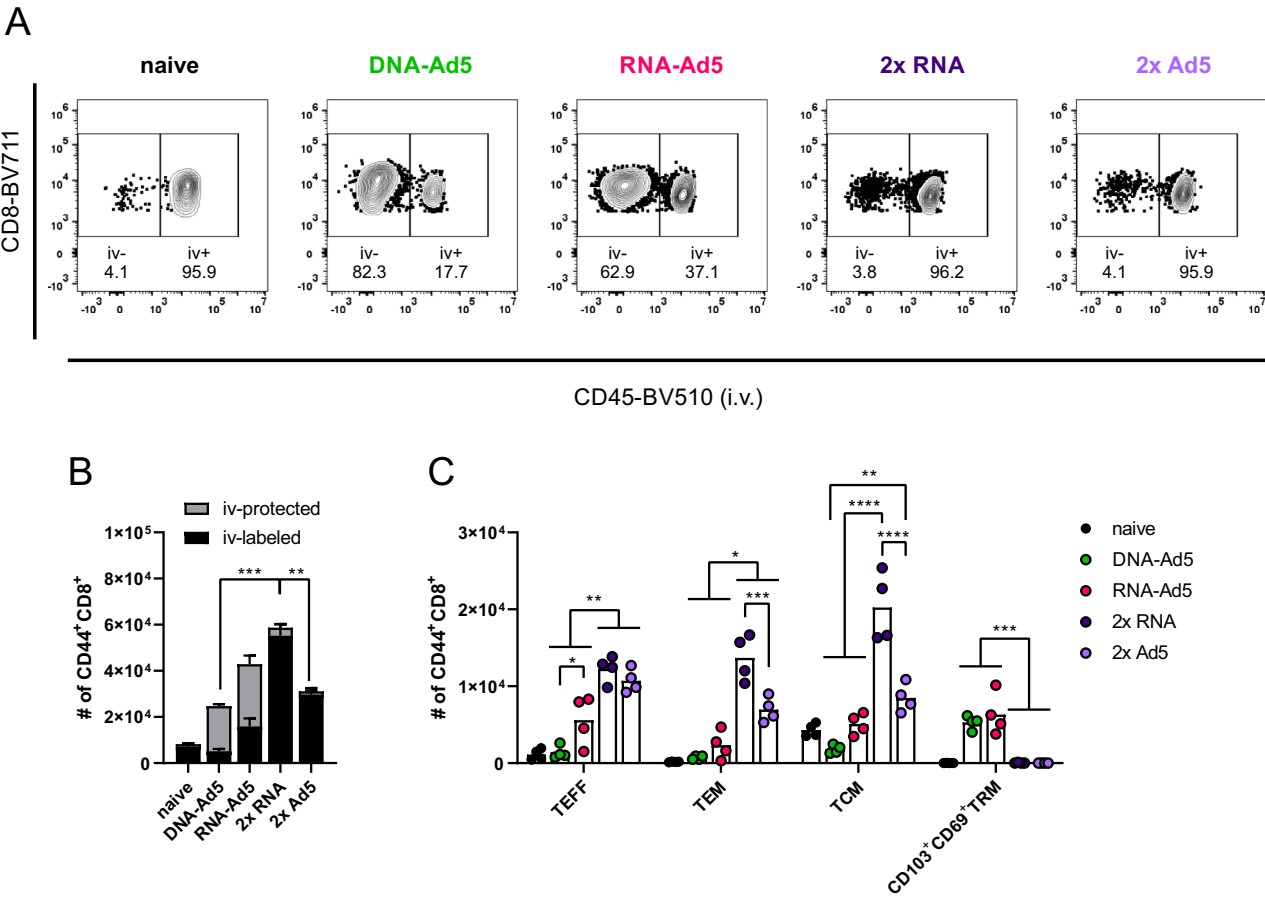

**Fig. 7 Circulating and tissue-resident memory T cell subsets in the lung.** C57BL/6 mice were vaccinated according to Fig. 5A. Antigen-experienced CD8$^+$ T cells were identified by CD44 staining and intravascular staining was used to differentiate between circulating (iv-labelled) and tissue-resident (iv-protected) memory cells. Representative contour plots are shown in (**A**). **B** The total number of CD44$^+$ CD8$^+$ with the relative contribution of iv− and iv+ cells are summarized for each group. **C** Within the iv-labelled CD44$^+$ CD8$^+$ population, effector T cells (T$_{EFF}$; CD127$^-$KLRG1$^+$), effector memory T cells (T$_{EM}$; CD127$^+$KLRG1$^+$), and central memory T cells (T$_{CM}$; CD127$^+$KLRG1$^-$CD69$^-$CD103$^-$) were defined. Within the iv-protected population, T$_{RM}$ cells were defined as KLRG1$^-$CD103$^+$CD69$^+$. The gating strategy is shown in Supplementary Fig. 2. Bars represent group means overlaid with individual data points; all groups $n = 4$. Data were analysed by one-way ANOVA followed by Tukey's multiple comparison test. Statistically significant differences are indicated only among the different vaccine groups; $p$ values indicate significant differences (*$p < 0.05$; **$p < 0.005$; ***$p < 0.0005$; ****$p < 0.0001$). Representative data from one out of three independent experiments with slightly different end time points are shown.

(Fig. 10F). Due to the nature of this challenge model, high viral RNA copy numbers were also detected in the brains of naïve animals (Supplementary Fig. 7). Although viral RNA was still detectable in the brains of most vaccinated animals, the copy numbers were reduced by 4–5 logs, and no significant differences among the vaccine groups could be seen.

Taken together, the mucosal boost strategies were able to fully prevent mortality and symptomatic disease upon experimental SARS-CoV-2 infection. The protective efficacy was equal to the current approved vaccination regimen consisting of two intramuscular injections of Comirnaty®.

## Discussion

The SARS-CoV-2 pandemic had and still has a deep impact on social, economic, and healthcare aspects of the world community. As a reaction, academic institutions, biotech companies, and regulatory agencies released safe and effective vaccines in an unprecedented speed. While early in the pandemic the vaccine efficacies of the respective vaccine schedules were in focus, the interest now shifts towards investigating immunogenicity and efficacy of mixed modality vaccinations, the maintenance of long-term immunity, and the protection against emerging variants. So far, the heterologous

combination of different vaccine modalities is mostly connected to a superior immunogenicity in preclinical[57,58] and clinical studies[59–62]. As now most countries with progressed vaccination campaigns discuss the employment of booster vaccinations, a possible next step might be the implementation of mucosal immunizations in order to harness the full potential of mucosal immunity at the entry port of SARS-CoV-2 infections.

To this end, the current study determined the immunogenicity and protective efficacy of mucosal boost vaccination with replication-incompetent adenoviral vectors after systemic prime immunizations with a DNA vaccine or an mRNA vaccine that is part of the current vaccine campaigns. The results clearly prove a potent induction of mucosal immune responses by these heterologous strategies not seen after purely systemic immunization schedules. Concomitantly, we observed comparable protective efficacies upon experimental infection among systemic immunization schedules and the heterologous mucosal boost strategies. These results should encourage the exploitation of mucosal booster immunizations as a non-traumatic vaccination modality able to induce strong mucosal immunity in addition to systemic responses. Although not discernible in the present study, this front-line immunity might further inhibit breakthrough infections and transmission risk.

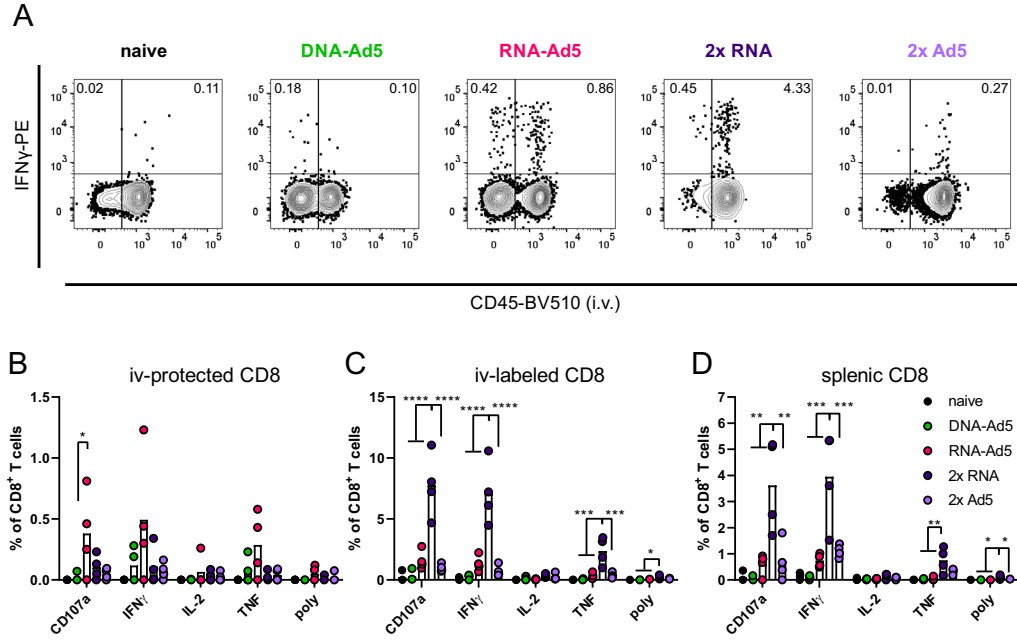

**Fig. 8 Spike-specific CD8⁺ T cell responses.** C57BL/6 mice were vaccinated according to Fig. 5A. Lung (**B** and **C**) and spleen homogenates (**D**) were restimulated with a peptide pool covering major parts of S. The responding CD8⁺ T cells were identified by intracellular staining for accumulated cytokines or staining for CD107a as degranulation marker. **A** Representative contour plots showing IFNγ production in iv+ and iv− lung CD8⁺ T cells. The gating strategy is shown in Supplementary Fig. 3. Bars represent group means overlaid with individual data points; all groups $n = 4$ (exception: $n = 3$ for DNA-Ad5 in **D**). Data were analysed by one-way ANOVA followed by Tukey's multiple comparison test. Statistically significant differences are indicated only among the different vaccine groups; $p$ values indicate significant differences (*$p < 0.05$; **$p < 0.005$; ***$p < 0.0005$; ****$p < 0.0001$). poly; polyfunctional T cell population positive for all assessed markers. Representative data from one out of three independent experiments with slightly different end time points are shown.

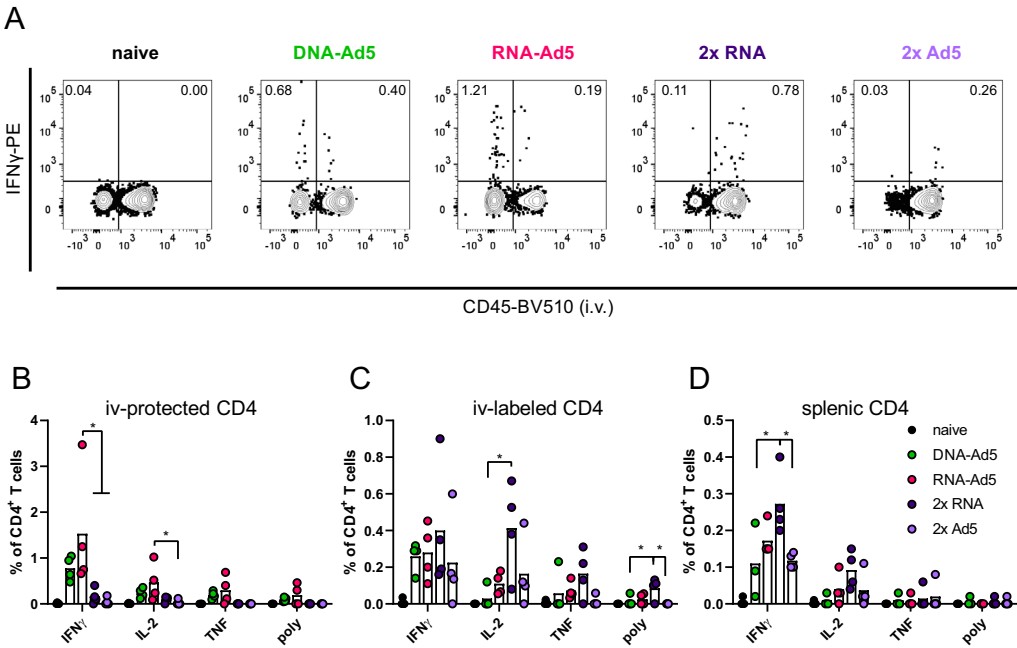

**Fig. 9 Spike-specific CD4⁺ T cell responses.** C57BL/6 mice were vaccinated according to Fig. 5A. Lung (**B** and **C**) and spleen homogenates (**D**) were restimulated with a peptide pool covering major parts of S. The responding CD4⁺ T cells were identified by intracellular staining for accumulated cytokines. **A** Representative contour plots showing IFNγ production in iv+ and iv− lung CD4⁺ T cells. The gating strategy is shown in Supplementary Fig. 3. Bars represent group means overlaid with individual data points; all groups $n = 4$ (exception: $n = 3$ for DNA-Ad5 in **D**). Data were analysed by one-way ANOVA followed by Tukey's multiple comparison test. Statistically significant differences are indicated only among the different vaccine groups; $p$ values indicate significant differences (*$p < 0.05$; **$p < 0.005$; ***$p < 0.0005$; ****$p < 0.0001$). poly; polyfunctional T cell population positive for all assessed markers. Representative data from one out of three independent experiments with slightly different end time points are shown.

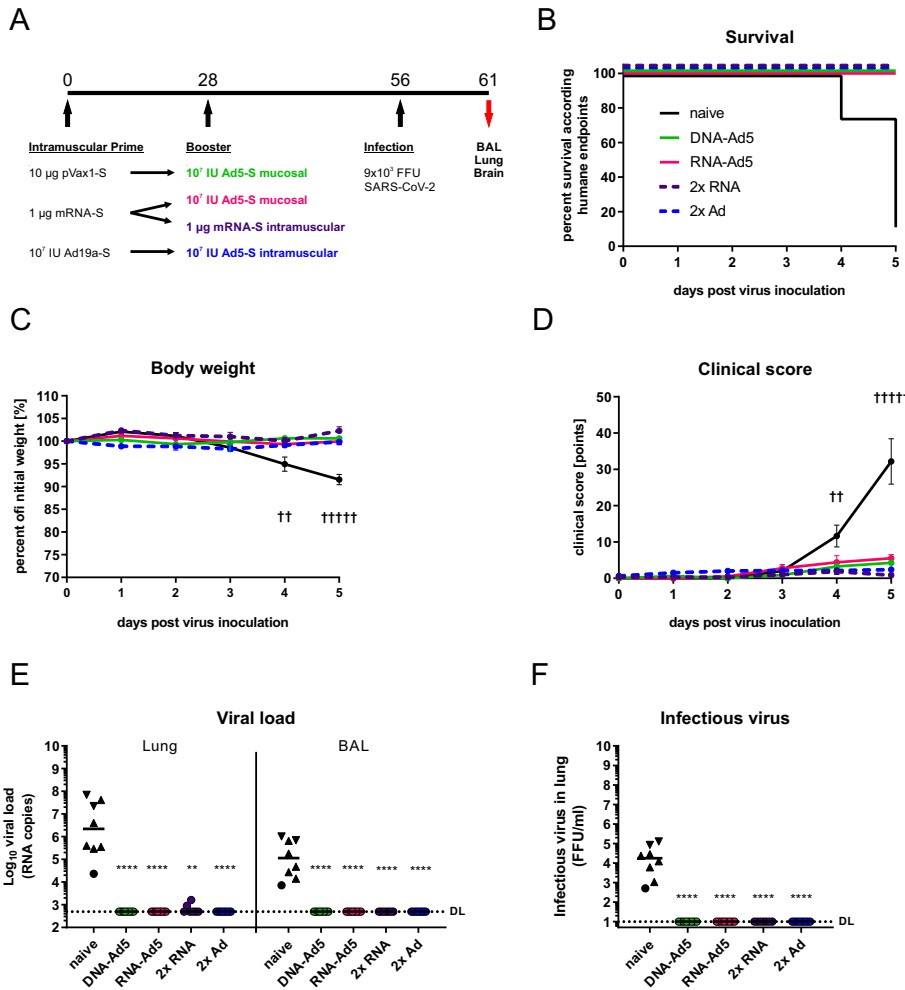

**Fig. 10 Protective efficacy against SARS-CoV-2 infection. A** K18-hACE2 mice (2x RNA $n = 7$, other groups $n = 8$) received an intramuscular prime immunization with the spike-encoding DNA (10 µg) followed by electroporation, Ad19a-S ($10^7$ infectious units), or the mRNA vaccine, Comirnaty® (1 µg). Mice from the heterologous prime-boost groups were boosted four weeks later intranasally or intramuscularly with Ad5-S ($10^7$ infectious units). The 2x RNA group received a second dose of mRNA (1 µg) intramuscularly. Four weeks after the boost immunization, mice were infected intranasally with $9 \times 10^3$ FFU SARS-CoV-2. All animals were monitored daily for survival (**B**), body weight (**C**), and clinical score (**D**). Curves in (**C**) and (**D**) represent group means with SEM. Animals reaching humane endpoints were euthanized and are marked by a cross at the respective time point. Viral RNA copy numbers were assessed in lung homogenates and BAL samples by qRT-PCR (**E**) and infectious virus was retrieved and titrated from lung homogenates (**F**). Data points shown represent viral copy number or virus titre of each animal with the median of each group, whereby circles indicate a survival of 5 days post infection and triangles indicates euthanized mouse according humane endpoints at day 4 (triangle pointing down) or day 5 (triangle pointing up). The dashed line indicates the lower limit of detection. Data were analysed by Kruskal–Wallis test (one-way ANOVA) and Dunn's Pairwise Multiple Comparison Procedures as post hoc test in comparison to PBS control; $p$ values indicate significant differences (*$p < 0.05$; **$p < 0.005$; ***$p < 0.0005$; ****$p < 0.0001$).

In the first part of the present study, we confirmed that a systemic prime with a plasmid DNA vaccine potentiates the immunogenicity of mucosally applied adenoviral vector vaccines[63]. Most probably, systemic memory cells induced by the priming expand during the recall response and are then recruited to the mucosal site to differentiate into tissue-resident memory cells as reported for $T_{RM}$ in the female reproductive tract[64]. This is an important finding since it clearly proves the suitability to implement mucosal immunizations into current SARS-CoV-2 vaccination schedules.

Similar to our observations, other preclinical studies imply significant protection against SARS-CoV-2 in mice, hamsters, ferrets, and non-human primates after an intranasal vaccination with adenoviral vector vaccines[47–49,51,65]. Hassan and colleagues showed that a single intranasal immunization with ChAd-SARS-CoV-2-S is superior to an intramuscular immunization with the same vaccine in regard to virus replication in the lower

respiratory tract[47]. They report similar findings in rhesus macaques as well, but in this animal model early virus replication in the upper respiratory tract was less affected by a mucosal vaccination[48]. Importantly, these studies did not assess heterologous prime-boost-strategies, which may increase the immunogenicity of the mucosal vaccination further and thereby might reduce early virus replication more effectively. Along these lines, the first data from a human clinical trial with an intranasal Ad5-based SARS-CoV-2 vaccine, Altimmune's AdCOVID, were disappointing and the development was discontinued[52]. Although safe and well tolerated, the vaccine did not demonstrate sufficient immunogenicity after one or two intranasal doses in previously unvaccinated individuals. The data from a very recent clinical phase I trial underline these findings by providing that serum antibody levels were lower after two intranasal doses of an Ad5-based SARS-CoV-2 vaccine than after one single intramuscular injection[53]. However, the combination of a mucosal booster

immunization with an intramuscular prime resulted in the highest levels of neutralising antibodies reported in that clinical trial. Unfortunately, mucosal immune responses were not assessed. These observations might support our notion of the potential need of pre-existing memory cells to maximize the immunogenicity of an intranasal immunization. While this might seem to complicate the use of nasal vaccines at first sight, one has to keep in mind that mass vaccination campaigns currently employ intramuscular immunizations in large parts of the community. Such heterologous vaccine strategies will finally result in a balanced response between systemic and mucosal immunity.

In most mucosal parameters observed, Ad19a was less immunogenic compared to Ad5, whereas systemic responses, especially CD4$^+$ T cell responses, were more efficiently induced. We reported this trend before and speculate that different tropisms of the viral vectors might account for that: Ad5 enters cells via binding to the coxsackie-and-adenovirus receptor (CAR), while Ad19a binds sialic acids and CD46 as entry receptors[66–69]. Since these molecules are differentially expressed on stromal and immune cells, this might be one aspect explaining the different local and systemic immune profiles. Importantly, also licensed adenoviral SARS-CoV-2 vaccines differ in their entry receptor usage. While AstraZeneca's Vaxzevria® (ChAdOx1) utilizes CAR as primary receptor, Janssen's Ad26.COV2.S enters cells via binding to CD46[70]. Thus, based on our observations, ChAdOx1 might be a good choice for mucosal immunizations, while Ad26 might be better suited for systemic vaccinations. Of note, the E1/E3 deletions to render adenoviral vectors replication-incompetent are similar among our Ad5/Ad19a vectors and the licensed Ad26 and ChAdOx1 vaccines.

In the second part of the present study, the impact of the systemic priming modality (mRNA/plasmid DNA) on the immunogenicity and efficacy of intranasal boost vaccinations with Ad5 was investigated and compared to two systemic immunizations with Ad5 or RNA. Humoral responses in the serum were largely comparable among all groups with the exception that two doses Ad5 provoked weaker responses. It is tempting to speculate that anti-vector immunity induced by the primary immunization might have dampened the effect of the homologous booster. This phenomenon is also discussed in the context of the lower vaccine efficacy in humans reported with two standard doses Vaxzevria® (ChAdOx1) compared to the low dose-standard dose schedule[5].

Mucosal antibody levels were higher in the groups having received a mucosal boost compared to the repeated systemic vaccination regimens. In regard to the levels of mucosal IgG, this trend was less pronounced as for mucosal IgA levels, presumably because serum IgG can be transudated into the respiratory lumen, whereas IgA is more stringently connected to a local immune reaction. Most importantly, the increased antibody responses in the mucosa also translated into more efficient virus neutralization by BAL samples. Only BAL samples from groups with mucosal vaccinations displayed neutralization of all tested VOCs. Although definitive evidence is currently missing, mucosal virus neutralization might be key to supress initial infections with SARS-CoV-2 and therefore minimize the risk of transmission to and by vaccinees. Interestingly, we observed distinct neutralization profiles between the DNA-Ad5 and RNA-Ad5 schemes probably originating from the use of different spike antigens. Thus, it is important to investigate the role of the prefusion conformation stabilization[71] regarding neutralization profiles in more detail.

An important advantage of intranasally administered genetic vaccines is the induction of $T_{RM}$ in the respiratory tract. In the present study, tissue-resident memory was exclusively established by mucosal vaccinations. This is congruent with published research showing that local antigen expression is essential for the development of respiratory $T_{RM}$[30,32,40]. Moreover, in combination with a mucosal boost, a priming immunization with mRNA provoked a broader cellular immunity compared to a plasmid DNA prime consisting of not only $T_{RM}$ in the lung but also of significant numbers of circulating memory T cells. We speculate that such comprehensive T cell immunity is more effective against breakthrough infections than having only circulating or only resident T cells. Although the chosen challenge model in K18-hACE2 mice did not allow to decipher different degrees of protection, it implies that heterologous prime-boost vaccinations with an intranasal component are at least as protective as the currently licensed vaccine schedules. To further investigate potential advantages of mucosal immune responses, upcoming studies must mimic the settings more closely that likely contribute to breakthrough infections in vaccinated individuals: age[72], comorbidities[73,74], waning immunity, and infection with less neutralization-sensitive variants like B.1.617.2[12–18]. However, experimental human challenge studies with a small number of participants might also illuminate this topic similar to challenge studies previously performed in the context of RSV[45,46] or Influenza[75,76].

Absolute or mechanistic correlates of protection against SARS-CoV-2 are not yet determined, although neutralizing antibody responses in sera were recently described to be predictive of protection against symptomatic infections[77,78]. However, limiting the initial infection rate by mucosal IgA and an early control of viral replication by local CD8$^+$ $T_{RM}$ would add another layer of protection, which may be underestimated so far. Furthermore, the rapid inhibition of local replication may result in reduced levels of pro-inflammatory cytokines that partially contribute to tissue damage and severe disease progression[79]. In face of the encouraging results from the mixed vaccine regimens using a mRNA vaccine first followed by adenoviral vector vaccines[59–62], it might be worthy to utilize an intranasal application for the viral vector boost immunization. This atraumatic, non-invasive application might also reduce the systemic side effects reported for the viral vector vaccines[80,81].

Finally, we demonstrated that the heterologous mRNA prime/intranasal Ad5 boost immunization is not inferior to the common gold standard of two intramuscular mRNA immunizations in regard to efficacy and additionally results in an unmatched mucosal immune response to SARS-CoV-2. Thus, this study provides the basis to pursue further efficacy studies in non-human primate models or even initiate clinical phase I studies using the currently available vaccines.

The induction of mucosal front-line immunity by heterologous vaccination strategies has the potential to mitigate current and future respiratory virus epidemics and pandemics. While maximizing the individual protection against breakthrough infections, it likely also decreases disease severity and the risk for virus transmission upon infections.

## Methods

**Ethics statement**. The study was approved by the Government of Lower Franconia, which nominated an external ethics committee that authorized the experiments. Studies were performed under the project license AZ 55.2.2-2532-2-1179. The infection experiments were approved by local authorities after review by an ethical commission (TVV21/20).

*Vaccines*. Codon-optimized sequences for the N or the spike S protein of SARS-CoV-2 were cloned into the pVAX1 expression plasmid (ThermoFisher) optimized for plasmid DNA vaccinations referred to as pVAX1-SARS2-N and pVAX1-SARS2-S[82]. The encoded S protein is the non-stabilized wildtype protein and based on the sequence of the initial Wuhan isolate (NCBI Reference Sequence: NC_045512.2; https://www.ncbi.nlm.nih.gov/nuccore/1798174254). Replication-deficient (ΔE1ΔE3) adenoviral vector vaccines based Ad5 or Ad19a/64 encoding the same antigens were produced as previously described[83] by Sirion Biotech

(Martinsried, Germany). In both vector systems, antigen expression is initiated from a CMV-immediate/early-1-promoter and a bovine BGH polyadenylation signal provides transcription termination. The mRNA vaccine Comirnaty® encodes the stabilized prefusion S protein and is described elsewhere[84].

**Immunizations.** Six to eight weeks old female BALB/cJRj or C57BL/6 mice were purchased from Janvier (Le Genest-Saint-Isle, France) and housed in individually ventilated cages in accordance with German law and institutional guidelines under specific pathogen-free (SPF) conditions, with constant temperature (20–24 °C) and humidity (45–65%) on a 12 h/12 h-light/dark cycle. Experimental and control animals were co-housed. The research staff was trained in animal care and handling in accordance to the FELASA and GV-SOLAS guidelines. For intramuscular immunizations, inhalative isoflurane anaesthesia was applied and the vaccines were injected in a volume of 30 μl PBS in the musculus gastrocnemius of each hind leg. In case of DNA immunizations, the injection was followed by electroporation as described previously[85]. Under general anaesthesia (100 mg/kg ketamine and 15 mg/kg xylazine), intranasal immunizations were performed by slowly pipetting a volume of 50 μl into one nostril containing the final vaccine dose. Blood was sampled from the retro-orbital sinus under light anaesthesia with inhaled isoflurane. For sampling BAL fluids, mice were killed and the lungs were rinsed twice with 1 ml cold PBS through the cannulated trachea.

**Antigen-specific antibody ELISA.** Spike S1, S2, and RBD antibody responses were analysed by ELISA. To this end, ELISA plates were coated with 100 ng of the respective peptide (RBD peptide provided by Diarect GmbH, Freiburg; S1 and S2 peptides kindly provided by Thomas Schumacher from Virion/Serion GmbH, Würzburg) in 100 μl carbonate buffer (50 mM carbonate/bicarbonate, pH 9.6) per well over night at 4 °C. Free binding sites were blocked with 5% skimmed milk in PBS-T (PBS containing 0.05% Tween-20) for 1 h at RT. BAL samples were diluted in 2% skimmed milk in PBS-T and incubated on the plate for one hour at RT. After three washing steps with 200 μl PBS-T, HRP-coupled anti-mouse IgA (dilution 1:5,000, Bethyl Laboratories) detection antibodies were added for 1 h at RT. Subsequently, the plates were washed seven times with PBS-T and after the addition of ECL solution, the signal was measured on a microplate luminometer (VICTOR X5, PerkinElmer) and analysed using PerkinElmer 2030 Manager software.

**FACS-based antibody analysis.** A modified version of our previously published serological assay was used[54], in which stably transduced HEK 293 T cells express the antigen of interest. To analyse quantities of antigen-specific antibodies, $5 \times 10^5$ HEK 293 T cells producing SARS-CoV-2 spike or nucleocapsid were incubated for 20 min at 4 °C with the respective biological sample diluted in 100 μl FACS-PBS (PBS with 0.5% BSA and 1 mM sodium azide) to bind to spike protein on the surface, or in 100 μl permeabilization buffer (0.5% saponin in FACS-PBS) to bind to intracellular nucleocapsid protein. After washing with 200 μl buffer, specific antibodies were bound with polyclonal anti-mouse Ig-FITC (1:300, 4 °C, 20 min incubation; BD Biosciences), anti-mouse IgG1-APC (1:300, clone X56, Biolegend), or anti-mouse IgG2a-FITC (1:300, clone R19-15, BD Biosciences). After further washing, samples were measured on an AttuneNxt (ThermoFisher) and analysed using FlowJo software (Tree Star Inc.).

**Virus neutralization assay.** Serial dilutions of sera and BALs were incubated with 2000 PFU of an early SARS-CoV-2 isolate (GISAID EPI ISL 406862 Germany/BavPat1/2020) in 100 μl OptiPro medium supplemented with 1x GlutaMAX (both Gibco) for 1 h at 37 °C. Subsequently, the mixture was added onto a confluent monolayer of Vero E6 cells (seeded the day before at $10^4$ cells per well in a 96-well plate). After 1 h, the mixture was removed from the cells and 100 μl OptiPro medium supplemented with 1x GlutaMAX (both Gibco) was added. After 24 h incubation at 37 °C and 5% CO2, cells were fixed with 100 μl 4% paraformaldehyde for 20 min at RT and permeabilized with 100 μl 0.5% Triton X-100 in PBS for 15 min at RT. After a blocking step with 100 μl 5% skimmed milk in PBS for 1 h at RT, cells were stained with purified immunoglobulins from a SARS-CoV-2 convalescent patient in 2% skimmed milk in PBS for 1 h at 4 °C. After three washing steps with 200 μl PBS, 100 μl of anti-human IgG FITC (1:200, Jackson Immunoresearch) were added diluted in 2% skimmed milk and incubated for 1 h at 4 °C in the dark. After another three washing steps with 200 μl PBS, plaques were counted with an ELISPOT reader and analysed using CTL Immunospot software (Cellular Technology limited BioSpot). Infected wells without serum were used as reference to determine the 75% plaque reduction neutralization titre (PRNT75).

**Pseudotype neutralization assay.** Neutralization of various spike variants was assessed with the help of spike-pseudotyped simian immunodeficiency virus particles as described before[86]. For the production of pseudotyped reporter particles, HEK293T cells were transfected with the SIV-based self-inactivating vector encoding luciferase (pGAE-LucW), the SIV-based packaging plasmid (pAdSIV3), and the respective spike variant-encoding plasmid[87–89]. For this purpose, $2 \times 10^7$ HEK293T cells were seeded the day before in a 175 cm² flask in Dulbecco's Modified Eagle's Medium (DMEM; Gibco) containing 10% FCS, 2 mM L-Glutamine, and 100 units/ml penicillin/streptomycin (D10 medium). The transfection mixture was prepared by mixing 20 μg of each plasmid with 180 μg polyethylenimine in 5 ml DMEM without additives. 15 min later, the mixture was

added to the cells. After 4–8 h incubation, the medium was exchanged to 25 ml DMEM containing 1.5% FCS, 2 mM L-Glutamine, and 100 units/ml penicillin/streptomycin. 72 h post-transfection, the supernatants containing the lentiviral particles were harvested, sterile filtrated (0.45 μm membrane), and stored at −20 °C. HEK293T-ACE2 cells stably expressing the human Angiotensin-converting enzyme 2 (ACE2) were transduced with the dilutions of the pseudo-types. The amount of lentiviral particles resulting in luciferase signals of $2–10 \times 10^4$ RLU/s were used for the latter assay.

For the assessment of pseudotype neutralization, HEK293T-ACE2 cells were seeded at $2 \times 10^4$ cells/well in 100 μl D10 in a 96well flat bottom plate. 24 h later, 60 μl of serial dilutions of the BAL samples were incubated with 60 μl lentiviral particles for 1 h at 37 °C. HEK293T cells were washed once with PBS and the particle-sample mix was added to the cells. 48 h later, medium was discarded and the cells lysed with 100 μl Bright Glo lysis buffer (Promega) for 15 min at 37 °C. Three minutes later, after the addition of 25 μl Bright Glo substrate (Promega), the luciferase signal was assessed on a microplate luminometer (VICTOR X5, PerkinElmer) and analysed using PerkinElmer 2030 Manager software. Neutralization titres are determined as the last reciprocal dilution that inhibits more than 75% of the luciferase signal measured in positive controls (inhibitors concentration 75%, IC75).

**T cell assays.** For the definition of circulatory and tissue-resident T cells, mice were injected with 3 μg anti-CD45-BV510 (clone 30-F11, Biolegend) intravenously and were euthanized 3 min later with an overdose of inhaled isoflurane. Spleens and lungs were harvested. The latter ones were cut into small pieces followed by incubation for 45 min at 37 °C with 500 units Collagenase D and 160 units DNase I in 2 ml R10 medium (RPMI 1640 supplemented with 10% FCS, 2 mM L-Glutamine, 10 mM HEPES, 50 μM β-mercaptoethanol and 1% penicillin/streptomycin). Digested lung tissues and spleens were mashed through a 70 μm cell strainer before the single cell suspensions were subjected to an ammonium-chloride-potassium lysis. One million splenocytes or 20% of the total lung cell suspension were plated per well in a 96-well round-bottom plate for in vitro restimulation and phenotype assays.

For the restimulation, samples were incubated for 6 h (or 24 h in case of Figs. 8, 9) in 200 μl R10 medium containing monensin (2 μM), anti-CD28 (1 μg/ml, eBioscience), anti-CD107a-FITC (1:200, clone eBio1D4B, eBioscience), and the respective SARS-CoV-2 peptide pool (0.6 nmol/peptide, S and N pools from Miltenyi Biotec, 130-126-701 and 130-126-699). Unstimulated samples were used for subtraction of background cytokine production. Cells were stained after the stimulation with anti-CD8a-Pacific blue (1:2000, clone 53-6.7, Biolegend), anti-CD4-PerCP (1:2000, clone RM4-5, eBioscience) and Fixable Viability Dye eFluor® 780 (1:4000, eBioscience) in FACS-PBS for 20 min at 4 °C. After fixation (2% paraformaldehyde, 20 min, 4 °C) and permeabilization (0.5% saponin in FACS-PBS, 10 min, 4 °C), cells were stained intracellularly with anti-IL-2-APC (1:300, clone JES6-5H4, Biolegend), anti-TNF-PECy7 (1:300, clone MPG-XT22, Biolegend), and anti-IFNγ-PE (1:300, clone XMG1.2, Biolegend). The gating strategy is shown in Supplementary Fig. 3.

For the phenotype analyses, cells were stained in FACS-PBS with anti-CD8-BV711 (1:300, clone 53-6.7, BioLegend), anti-CD4-BV605 (1:1000, clone RM4-5, BioLegend), anti-CD127-FITC (1:500, clone A7R34, BioLegend), anti-CD69-PerCP-Cy5.5 (1:300, clone H1.2F3, BioLegend), anti-CD103-PE (1:200, clone 2E7, eBioscience), anti-KLRG1-PE-Cy7 (1:300, clone 2F1, eBioscience), anti-CD44-APC (1:5000, clone IM7, BioLegend), and Fixable Viability Dye eFluor® 780 (1:4000, eBioscience). Data were acquired on an AttuneNxt (ThermoFisher) or on a LSRII (BD Biosciences) and analysed using BD FACS DIVA, ThermoFisher AttuneNext, and FlowJo™ software (Tree Star Inc.). The gating strategy is shown in Supplementary Fig. 2.

**SARS-CoV-2 infection model.** The infection experiments were approved by local authorities after review by an ethical commission (TVV 21/20). Eleven weeks old, female K18-hACE2 mice (stock # 034860, Jackson Laboratory, Bar Harbor, USA) were immunized as described before and infected four weeks after the boost immunization intranasally with $9 \times 10^3$ focus-forming units (FFU) of the SARS-CoV-2 strain BavPat1 in a total volume of 50 μl under light anaesthesia with inhaled isoflurane. Animals were monitored daily for body weight and clinical score. The following parameters were evaluated in the scoring system: weight loss and body posture (0–20 points), general conditions including the appearance of fur and eye closure (0–20 points), reduced activity and general behaviour changes (0–20 points), and limb paralysis (0–20 points). Mice were euthanized at day 5 after infection or earlier if a cumulative clinical score of 20 or more was reached by a final blood draw under deep narcosis. After euthanasia, the lungs were filled with 800 μl PBS and the left lung was tied off. The BAL of the right lung was taken and repeated with two more washes each with 400 μl. The right lungs as well as the right hemispheres of the brains were homogenized in 1 ml PBS using a gentle-MACS Octo Dissociator (Miltenyi Biotec) and viral RNA was isolated from 140 μl cleared homogenate or BAL fluid using QIAamp Viral RNA Mini Kit (Qiagen). RT-qPCR reactions were performed using TaqMan® Fast Virus 1-Step Master Mix (ThermoFisher) and 5 μl of isolated RNA as a template to detect a 132 bp sequence in the ORF1b/NSP14. Primer and probe sequences were as follows: forward primer, 3′-TGGGGYTTTACRGGTAACCT-5′; reverse primer,

AACRCGCTTAACAAAGCACTC; probe, 3′-FAM-TAGTTGTGATGCWAT-CATGACTAG-TAMRA-5′. Synthetic SARS-CoV-2-RNA (Twist Bioscience) was used as a quantitative standard to obtain viral copy numbers [90]. For the detection of infectious virus in BAL and the lung, Vero E6 cells were seeded at $2 \times 10^4$ cells/well in a 96-well plate in 200 µl of D10 for confluent monolayer 24 h prior to infection. After medium change to D10, a two-fold-serial dilution of BALs or lung homogenates were applied to the cells for 3 h. After replacing the supernatant with overlay medium (DMEM with 1% methyl cellulose, 2% FBS and 1% penicillin/streptomycin), cells were incubated for further 27 h. SARS-CoV-2 infected cells were visualized using SARS-CoV-2 S-protein specific immunochemistry staining with anti-SARS-CoV-2 spike glycoprotein S1 antibody (1:2000; Abcam) and an anti-human-IgG HRP detection antibody (1:1000, Dianova)[91].

*Statistical analyses.* Results are shown as mean ± SEM or as median ± interquartile range except it is described differently. Statistical analyses were performed with Prism 8.0 (GraphPad Software, Inc.). A $p < 0.05$ was considered to be statistically significant. For reasons of clarity, significances are only shown among the vaccine groups.

**Reporting summary**. Further information on research design is available in the Nature Research Reporting Summary linked to this article.

## Data availability

The raw numbers for charts and graphs are available in the Source Data file whenever possible. The following sequences are publicly accessible: NCBI Reference Sequence NC_045512.2 (https://www.ncbi.nlm.nih.gov/nuccore/1798174254) and GISAID EPI ISL 406862 Germany/BavPat1/2020. This paper does not report original code. Source data are provided with this paper.

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

## Acknowledgements

We would like to thank Drew Hannaman (Ichor Medical Systems, Inc.) for providing the TriGrid electrode array for DNA electroporation, Anne-Kathrin Donner for excellent technical assistance, and the group of Dr. Jasmin Fertey (Fraunhofer IZI, Leipzig, Germany) for providing viral stocks for mouse infections. Moreover, we kindly thank Thomas Schumacher (Virion/Serion GmbH, Würzburg, Germany) and Diarect GmbH (Freiburg, Germany) for providing the Spike peptides. The ELISPOT Analyzer was obtained with financial support from Fondation Dormeur, Vaduz. This work was supported within the FOR-COVID project funded by the Bavarian State ministry for Science and the Arts (to M.T. and K.Ü.). Further support was provided by B-FAST, a BMBF-funded project of the Netzwerk Universitätsmedizin (NaFoUniMedCovid19; FKZ: 01KX2021; to K.Ü.) and funds from the Deutsche Forschungsgemeinschaft (DFG) through the research training group RTG 2504 (project number: 401821119, to A.V.A, M.T., K.Ü.). S.P. acknowledges funding by BMBF (01KI2006D, 01KI20328A, 01KI20396, 01KX2021), the Ministry for Science and Culture of Lower Saxony (14-76103-184, MWK HZI COVID-19) DFG (PO 716/11-1, PO 716/14-1).

## Author contributions

D.L., T.G., K.Ü., and M.T. conceived and designed the study. D.L., J.F., J.W., A.V.A., V.E., N.U., L.I., A.S., F.O., A.S.P., P.I., and K.F. collected the data. D.L., J.F., J.W., A.V.A., V.E., N.U., L.I., T.G., and M.T. performed the analysis. S.M.-S., A.C., A.E., M.H., S.P., C.P., T.W., and C.T. contributed critical reagents. D.L. and M.T. drafted the manuscript, which was then critically reviewed and approved by all co-authors.

## Funding

## Competing interests

C.T. is founder and shareholder of SIRION Biotech GmbH. The other authors declare no competing interests.
