## [Peer Review File · Nature Communications]

REVIEWERS' COMMENTS

Reviewer #1 (Remarks to the Author):

In this manuscript, Lapuente and colleagues explore preclinical administration of intramuscular DNA and RNA COVID vaccines in combination with an intranasal adenovirus vectored vaccine. The main conclusion is that the administration of a mRNA vaccine (Cominarty, Biontech/Pfizer) followed by a boost of intranasal Ad5 boost, is immunogenic and effective against SARS-CoV-2 infection in a murine model of infection. They propose that this study serve as basis to develop non-human primates and human clinical trials testing this combination.

Major comments:

1. Considering that a clinical trial recently published in Lancet Infectious Diseases (Shipo Wu et al 2021) demonstrated promising results combining both intramuscular and intranasal adenovirus vectored vaccines, and the evidence in multiple papers of strong immune responses in humans to heterologous Ad vector + RNA COVID-19 vaccines, the data on preclinical stage reported here seems more appropriate for a more specialized journal at this point.
2. Lines 167-168, it does not seem that anti-Spike IgG levels in response to homologous RNA vaccination is superior to the other groups. For the graphs, individual dots could be used, even if combined with bars, to clarify questions like this.
3. It will be valuable to discuss how the data presented by the authors compare to Michal Diamond's lab previous reports (Hassan et al Cell; Hassan et al Cell Reports)

Minor comments:

4. Lines 118-119 - the authors mentioned that predominant polarization is not indicated for any vaccine group, but it looks like IgG2 is generally higher for all groups. Absence of direct comparison between IgG1 and IgG2 impairs any conclusion in my opinion.
5. Figure 6 - have the authors compared the response to variants with that from WT?
6. In the Methods there is no mention of how IgG and subtypes were determined and is not clear why RLU was used to measure IgA.

Reviewer #2 (Remarks to the Author):

The manuscript by Lapuente et al. is an examination of the comparative immunity in mice immunized by DNA, mRNA, and adenovirus strategies, including a prime-boost strategy involving an intranasal/mucosal boost with adenovirus. The studies address a number of timely and interesting issues of relevance to the larger questions of vaccination strategies against mucosal virus infections. The methods cover most of the technical questions relating to the systemic and mucosal immunity generated by the vaccination strategies, and the authors draw fair conclusions from the data without over-interpretation.

This is an important report that provides useful insights into the impact of the prime-boost mucosal vaccination while also raising a number of interesting questions. First, the studies, perhaps not completely surprisingly, show that the mucosal boost strategy provides a strong lung tissue T and B cell response with particularly strong IgA titers with virus neutralizing activity. This confirms the dramatic difference between systemic vaccination and mucosal boost strategies with regard to mucosal IgA production, and will re-activate the debate on whether mucosal IgA as first-line protection against infection is a more effective strategy than the prevailing practice of systemic immunization against viruses such as influenza.

This reader's conclusion from these studies is that the debate will remain open, as the effective protection from actual infection is similar between systemic and mucosal immunity, presumably because of the importance in T cell and IgG immunity even after an initial mucosal infection is presumed to occur. So the basic biology question of whether mucosal IgA is a critical requirement for protection from mucosal virus infection may remain open as a policy debate in public immunization strategies. It is possible that the model is not able to test for various conditions where the mucosal IgA may show a more critical importance, but it will be interesting to watch this debate continue. A related question, briefly touched on in the discussion, is whether an intranasal boost is simply more easily administered in clinical practice, and therefore may have other more practical arguments in favor of this strategy even in the absence of strong efficacy arguments.

Reviewer #3 (Remarks to the Author):

This paper by Lapuente et al provides clear evidence that a prime-boost vaccination strategy against SARS CoV2 that includes an intranasal boost generates mucosal immunity in the lung. In contrast, the current intramuscular schedules induce only systemic immunity, which may not be useful in preventing virus infection and spread. While not necessarily surprising, the results are important evidence to support the need for mucosal vaccination as a strategy for driving effective population immunity against Covid-19. The data are generally clear and presented logically, and my only general concern is that the results are entirely descriptive in nature, with no exploration of more fundamental aspects. Other specific comments:

- 1) The title of the paper rather overstates the findings, as it focuses on the mRNA vector as being the effective vector. However the data show that the Ad-based vectors are also generally useful in priming mucosal responses and the differences between these and mRNA are not consistent. The title should be modified to reflect this.
- 2) No information is provided on how often experiments were repeated.
- 3) The authors might want to comment on how the Ad vectors used here are similar or different to those currently used in the commercial vaccines of this kind.
- 4) The mouse model of infection is potentially interesting, but it has to be acknowledged that the results do not correlate well with the findings of the specific immune responses. Essentially it shows that any form of vaccination is protective in this system, which may therefore be too crude to reveal any effects of mucosal immunisation. The authors might consider omitting these data or transferring them to a Supplementary figure.

Point-by-point response to the reviewer's comments

Reviewer #1:

In this manuscript, Lapuente and colleagues explore preclinical administration of intramuscular DNA and RNA COVID vaccines in combination with an intranasal adenovirus vectored vaccine. The main conclusion is that the administration of a mRNA vaccine (Cominarty, Biontech/Pfizer) followed by a boost of intranasal Ad5 boost, is immunogenic and effective against SARS-CoV-2 infection in a murine model of infection. They propose that this study serve as basis to develop non-human primates and human clinical trials testing this combination.

Major comments:

1. Considering that a clinical trial recently published in Lancet Infectious Diseases (Shipo Wu et al 2021) demonstrated promising results combining both intramuscular and intranasal adenovirus vectored vaccines, and the evidence in multiple papers of strong immune responses in humans to heterologous Ad vector + RNA COVID-19 vaccines, the data on preclinical stage reported here seems more appropriate for a more specialized journal at this point.

Response:

We agree that there is some progress in the development of vaccine strategies incorporating a mucosal boost component. However, the specific combination of systemic RNA and mucosal AdV booster immunization has not be described before. In light of the current licensed vaccination programs, these data might provide the rational to use licensed AdV-based vaccines, like Vaxzevria or COVID-19 vaccine Janssen, in people who had been vaccinated with mRNA-based vaccines before to strengthen the local immune response. Therefore, we claim substantial novelty for this data set complementary to the existing and growing literature on heterologous prime-boost immunizations.

2. Lines 167-168, it does not seem that anti-Spike IgG levels in response to homologous RNA vaccination is superior to the other groups. For the graphs, individual dots could be used, even if combined with bars, to clarify questions like this.

Response:

According to the journals policy and the reviewer's suggestion, we included now individual dots in each bar graph to visualize also the variance within the groups.

3. It will be valuable to discuss how the data presented by the authors compare to Michal Diamond's lab previous reports (Hassan et al Cell; Hassan et al Cell Reports)

Response:

We already cited and discussed the mentioned papers in the original manuscript, but we added an additional statement in the revised manuscript. However, our data add complementary and new knowledge to the existing reports by combining the mucosal vaccination with the systemic priming. This is important since a current clinical trial based solely on intranasal AdV immunization was stopped because it shows low immunogenicity (Altimmune)

Minor comments:

4. Lines 118-119 - the authors mentioned that predominant polarization is not indicated for any

vaccine group, but it looks like IgG2 is generally higher for all groups. Absence of direct comparison between IgG1 and IgG2 impairs any conclusion in my opinion.

Response

We agree with the reviewer that this statement is misleading since we did not quantify the different IgG subclasses. We rephrased this part in the revised manuscript.

5. Figure 6 - have the authors compared the response to variants with that from WT?

Response

We compared it to the neutralization against pseudotypes carrying the S protein of the D614G variant which emerged very rapidly in the course of the pandemic and is closely related to the WT. The responses were comparable to the neutralization of the B.1.1.7 variant. The data is included in the revised Fig.6

6. In the Methods there is no mention of how IgG and subtypes were determined and is not clear why RLU was used to measure IgA.

Response

In the methods, an antigen-specific ELISA is described for the measurement of IgA using HRP-coupled secondary reagents and chemiluminescence as readout. Therefore, we provide RLUs as units.

IgG and IgG subtypes were determined in the FACS-based antibody assay described in the methods section.

Reviewer #2

The manuscript by Lapuente et al. is an examination of the comparative immunity in mice immunized by DNA, mRNA, and adenovirus strategies, including a prime-boost strategy involving an intranasal/mucosal boost with adenovirus. The studies address a number of timely and interesting issues of relevance to the larger questions of vaccination strategies against mucosal virus infections. The methods cover most of the technical questions relating to the systemic and mucosal immunity generated by the vaccination strategies, and the authors draw fair conclusions from the data without over-interpretation.

This is an important report that provides useful insights into the impact of the prime-boost mucosal vaccination while also raising a number of interesting questions. First, the studies, perhaps not completely surprisingly, show that the mucosal boost strategy provides a strong lung tissue T and B cell response with particularly strong IgA titers with virus neutralizing activity. This confirms the dramatic difference between systemic vaccination and mucosal boost strategies with regard to mucosal IgA production, and will re-activate the debate on whether mucosal IgA as first-line protection against infection is a more effective strategy than the prevailing practice of systemic immunization against viruses such as influenza.

This reader's conclusion from these studies is that the debate will remain open, as the effective protection from actual infection is similar between systemic and mucosal immunity, presumably because of the importance in T cell and IgG immunity even after an initial mucosal infection is presumed to occur. So the basic biology question of whether mucosal IgA is a critical requirement for protection from mucosal virus infection may remain open as a policy debate in public immunization

strategies. It is possible that the model is not able to test for various conditions where the mucosal IgA may show a more critical importance, but it will be interesting to watch this debate continue. A related question, briefly touched on in the discussion, is whether an intranasal boost is simply more easily administered in clinical practice, and therefore may have other more practical arguments in favor of this strategy even in the absence of strong efficacy arguments.

Response:

We thank the reviewer for the overall positive feedback on our study. We totally agree with the limitations in the conclusions of our infection model and that there are still open questions. But as the reviewer states in the report, we are convinced that this study delivers important knowledge for the general management of vaccinations against respiratory viruses.

Reviewer #3:

This paper by Lapuente et al provides clear evidence that a prime-boost vaccination strategy against SARS CoV2 that includes an intranasal boost generates mucosal immunity in the lung. In contrast, the current intramuscular schedules induce only systemic immunity, which may not be useful in preventing virus infection and spread. While not necessarily surprising, the results are important evidence to support the need for mucosal vaccination as a strategy for driving effective population immunity against Covid-19. The data are generally clear and presented logically, and my only general concern is that the results are entirely descriptive in nature, with no exploration of more fundamental aspects.

Response:

We thank the reviewer for the overall positive feedback on our study.

Other specific comments:

1) The title of the paper rather overstates the findings, as it focuses on the mRNA vector as being the effective vector. However the data show that the Ad-based vectors are also generally useful in priming mucosal responses and the differences between these and mRNA are not consistent. The title should be modified to reflect this.

Response:

Thank you for this helpful comment. We adjusted the title in the revised manuscript

2) No information is provided on how often experiments were repeated.

Response:

The information is provided in the revised manuscript and in the reporting summary which will be published alongside the manuscript.

3) The authors might want to comment on how the Ad vectors used here are similar or different to those currently used in the commercial vaccines of this kind.

Response:

We added respective information to the discussion section.

4) The mouse model of infection is potentially interesting, but it has to be acknowledged that the results do not correlate well with the findings of the specific immune responses. Essentially it shows that any form of vaccination is protective in this system, which may therefore be too crude to reveal any effects of mucosal immunisation. The authors might consider omitting these data or transferring them to a Supplementary figure.

Response:

We totally agree that this model was not suitable to demonstrate superiority of the RNA/AdV compared to the solely systemic immunization. However, we think it is as important to demonstrate non-inferiority which we could clearly show. Given the higher systemic antibody and T-cell responses in the 2xRNA group, it would have been also possible that these groups were better protected. Since a real correlate of protection is not known, non-inferiority to the licensed effective vaccines would be the first step. Therefore, we think the challenge data are important and should be part of the main manuscript. However, we rephrased the discussion of the limitations of this part for more clarity.